# Characterization and Tribological Behavior of Electroless-Deposited Ni-P-PTFE Films on NBR Substrates for Dynamic Contact Applications

Beatriz Vasconcelos [1,2], Ricardo Serra [1], João Oliveira [1] and Carlos Fonseca [2,3,*]

1   CEMMPRE—Centre for Mechanical Engineering, Materials and Processes, Department of Mechanical Engineering, University of Coimbra, Rua Luís Reis Santos, 3030-788 Coimbra, Portugal
2   Department of Metallurgical and Materials Engineering, Faculty of Engineering, University of Porto, Rua Dr. Roberto Frias, s/n, 4200-465 Porto, Portugal
3   LAETA/INEGI-Institute of Science and Innovation in Mechanical and Industrial Engineering, Rua Dr. Roberto Frias, 4200-465 Porto, Portugal
*   Correspondence: cfonseca@fe.up.pt

**Abstract:** The use of rubber in dynamic contacts often results in severe degradation and wear of the rubber surface, which is why dynamic rubber seal contacts are usually oil lubricated to ensure their functionality. However, the increasing demand for more convenient and environmentally friendly sealing solutions has prompted the development of dry low-friction rubber coatings. In this work, and for the first time, Ni-P and polytetrafluoroethylene (PTFE) particles were co-deposited by electroless plating on Nitrile Butadiene Rubber (NBR), as a low-cost solution to improve the NBR tribological behavior. A cationic surfactant, cetyltrimethylammonium bromide (CTAB), was added to the plating bath to ensure a homogeneous and efficient incorporation of PTFE into the Ni-P. The optimized PTFE incorporation reached 6.8%, and the composite coating adhesion to NBR was 20% higher than that of nickel-phosphorous (Ni-P) films. The tribological properties of the coatings evaluated by pin-on-disk tests showed a marginal decrease in the coefficient of friction (CoF) (10%, 1 N load), compared to that of Ni-P. However, the tested PTFE-based coatings displayed significantly smoother surfaces with less debris and cracks, clearly demonstrating the benefits of the PTFE in terms of wear resistance for loads up to 5 N.

**Keywords:** electroless plating; Ni-P-PTFE; co-deposition; coefficient of friction; NBR





## 1. Introduction

Elastomers are among the most used materials in applications that demand large elastic deformation, impact resistance, damping and flexibility. In the case of rubber, further attractive properties such as low weight, ease of manufacture and low production costs have contributed to its application in dynamic contacts and/or when sealant properties are also required, such as O-rings for zoom cameras, water pumps and some automobile components [1]. However, rubber lacks the wear resistance of other competing materials, and its use is limited by the large CoF when sliding against other engineering materials [2,3]. Thus, the use of rubber in dynamic contacts yields severe degradation and wear of the sliding surfaces, often forcing to component replacement. The easiest and long-used solution is to lubricate the rubber seal contact to minimize the wear and ensure long-term functionality. However, the risk of contaminations and the increasing demand for more environmentally friendly and straightforward solutions, i.e., lubrication-free protection of rubber components, has accelerated the development of new alternatives, such as the deposition of coatings over rubber surfaces [4].

So far, the most explored solutions for elastomer surface functionalization for dynamic contacts are the diamond-like carbon (DLC) coatings [4]. Despite the good initial

results, especially in lowering the CoF by 50%–80% [4–10], the deposition of DLC coatings, either by chemical vapor deposition (CVD) or physical vapor deposition (PVD) methods, faces several problems such as deficient adhesion and the temperature sensitivity of elastomers [4,5,7,8,11,12]. On the other hand, both deposition methods rely on the use of heavy and expensive equipment, including high vacuum systems, which significantly increase the production costs of an otherwise low-cost material.

An alternative, easier and less costly solution is the rubber deposition of a thin metallic film with intrinsic lubricant properties, such as Ni-P, using a less expensive deposition method such as electroless plating. Indeed, Ni-P coatings are frequently used for corrosion protection and for enhancing the tribological properties of metallic parts used as tools and the inner parts of molds [13,14].

In a previous work [15] the authors already succeeded in decreasing the CoF of NBR by 50% through a green surface modification with a polydopamine (PDA) coating, followed by electroless Ni-P plating. A possible strategy to further improve Ni-P coatings tribological performance is to co-deposit a lubricating filler, e.g., graphite, molybdenum disulfide, PTFE, or ceramic nanoparticles such as silicon nitride, alumina or silicon carbide [14]. However, ceramic particles were avoided since they may wear the material in contact with rubber, compromising the sealing, and PTFE was chosen among the solid lubricants. PTFE is a low friction and corrosion resistant polymer [14,16,17], known for being self-lubricant [18] and its co-deposition with Ni-P has been shown to significantly decrease the wear rate and CoF of metallic substrates [19–23]. However, as far as the authors are aware, Ni-P-PTFE composite coatings have not been investigated for the protection of non-metallic surfaces, particularly rubber surfaces.

Ni-P-PTFE composites are only useful for tribological applications if there is an adequate incorporation and dispersion of the PTFE particles in the metallic Ni-P matrix [19]. In that regard, the use of surfactants is critical and, additionally, requirements of bath stability, agitation, particle size and concentration of PTFE particles as well as bath composition are also important. The dynamics and co-dependence of such great number of variables have been investigated in several studies [24–28] resulting in the following conclusions: (i) the PTFE particles net charge should be positive, for which a surfactant should be used; (ii) surfactants must have some level of reactivity for their reduction on the cathode in order to improve Ni-P adhesion; (iii) there is an optimum ratio of surfactant to PTFE concentration for a maximum incorporation of PTFE with good particle distribution. Conditions (i) and (ii) are provided by cationic surfactants, usually of the fluoroalkylated class, FC-types, or CTAB. Condition (iii) should be optimized by the user in terms of bath composition, pH, and temperature.

The main aim of the present work was to improve the tribological properties of Ni-P films previously deposited over NBR substrates though co-deposition of PTFE by electroless plating. Electroless Ni-P-PTFE deposition was performed on a pre-modified NBR substrate using CTAB as the surfactant. The PTFE dispersion in the coating was assessed by scanning electron microscopy (SEM) coupled with energy dispersive spectroscopy (EDS) analysis (SEM/EDS), through measurement of the fluorine atomic content on the surface. Gravimetric analysis was used to measure the plating rate and coating's adhesion was evaluated by the direct pull-off (DPO) method. The tribological properties of the coatings were evaluated by pin-on-disc tests.

## 2. Materials and Methods

### 2.1. Surface Pre-Treatment and Activation

Samples sized $35 \times 12$ mm$^2$ were cut from an oil resistant NBR sheet with 3 mm thickness, shore hardness of 70 and acrylonitrile (ACN) content in the range 20%–30% (Munsch & Co./Polymer Trade Manufacturing Ltd., Bollington, UK). Both sides of the sample were abraded to remove a surface layer with approximately 0.5 mm, followed by ultrasonication in boiled distilled water for 30 min.

The cleaned samples were pre-treated as described in [15], in the following two steps: (i) chlorination and (ii) polydopamine (PDA) polymerization and reduction. Briefly, NBR samples were immersed in an aqueous chlorination solution of 2:1 NaOCl:HCl (NaOCl, 6%–14% active chlorine, EMPLURA®, MilliporeSigma, MA, USA; HCl, analytical reagent, VWR International Ltd., Leicestershire, UK) in 100 parts by volume of water for 3 min. After neutralization, PDA polymerization was carried out in a dopamine (DA) solution with a concentration of 2.0 g/L, using $CuSO_4$ (ACS reagent $\geq$ 98%, Sigma-Aldrich, Lisbon, Portugal) and $H_2O_2$ (30%, stabilized, for synthesis, Sigma-Aldrich) to accelerate the deposition in concentrations of 1.9 g/L and 0.5 mL/L, respectively. This was followed by catalytic activation with a 70 mM sodium borohydride solution (reagent grade, Riedel-de Haën, Sigma-Aldrich, Darmstadt, Germany) for 30 min. Between each step, the samples were thoroughly rinsed with distilled water. In the end, the samples are denominated reduced polydopamine-coated NBR (rPDA-NBR).

### 2.2. Electroless Ni-P-PTFE Alloy Plating

A 60% wt PTFE dispersion in water (Ø 0.05–0.5 μm particle size, Sigma-Aldrich, Germany) was used as the PTFE source for the composite coating. Electroless plating was carried out in a home-made Ni-P plating bath, and a cationic surfactant, CTAB (for synthesis, Sigma-Aldrich, Darmstadt, Germany), was used to disperse the PTFE particles. The plating conditions of concentration, temperature, and pH are described in Table 1, where a photo of a plated sample is also shown. The volume of the plating bath was 100 mL. All of the Ni-P-PTFE were obtained by first pre-depositing a thin Ni-P layer, by immersion in the plating bath, absent of PTFE and CTAB, for 15 min. Then, the PTFE dispersion, previously stirred in distilled water for 30 min, and a known amount of CTAB were added to the bath and plating proceeded for 1 h. After plating, the samples were taken out of the bath, rinsed with distilled water, and left to air-dry overnight in a fume hood.

**Table 1.** Plating bath conditions. An example of a plated sample can be seen in the table.

| Composition and Operating Conditions | |
| --- | --- |
| $NiSO_4 \cdot 6H_2O$ (g/L) | 40 |
| $C_6H_5Na_3O_7 \cdot 2H_2O$ (g/L) | 20 |
| $NaH_2PO_2 \cdot H_2O$ (g/L) | 40 |
| $C_2H_6BN$ (DMAB, g/L) | 2 |
| PTFE (mL/L) | 0, 5, 7, 10 |
| CTAB (mg/L) | 0, 200, 500, 800 |
| Temperature (°C) | 60 $\pm$ 3 |
| pH (adjusted with 3% $H_2SO_4$) | 5.0 $\pm$ 0.2 |
| Duration (min) | 60 |
| $NiSO_4 \cdot 6H_2O$ (g/L) | 40 |
| $C_6H_5Na_3O_7 \cdot 2H_2O$ (g/L) | 20 |

### 2.3. Characterization

2.3.1. Wettability and Chemical Composition

Contact angle (CA) of ultrapure water in various surfaces was calculated with the Young–Laplace approximation for 6 measurements per sample using a Theta Lite (TL100) Optical Tensiometer equipment (Biolin Scientific, Gothenburg, Sweden), at 5 frames per second. EDS (Energy dispersive X-ray spectroscopy, Genesis X4M, EDAX, Amtek, Tomball, TX, USA) was used for elemental analysis and mapping. The results obtained for fluorine content were used to estimate the percentage of PTFE incorporated in the samples.

### 2.3.2. Morphology and Crystalline Structure

The morphology of the samples was characterized by scanning electron microscopy (SEM, Quanta 400FEG ESEM, FEI company, Hillsboro, OR, USA) coupled with an EDAX Genesis X4M system for energy dispersive X-ray spectroscopy (EDS).

The crystallinity was evaluated by XRD in a PANalytical X'Pert PRO MPD equipment (Malvern Panalytical, Malvern, UK) using Cu Kα radiation (45 kV and 40 mA) with a parallel beam in θ–2θ geometry. A PIXcel detector (Malvern Panalytical, Malvern, UK) in receiving slit mode was used for X-ray collection.

### 2.3.3. Mechanical Properties

Quantification of the force at which detachment of the films occu, was performed through the direct pull-off (DPO) method described in the ASTM D4541 standard [29]. Epoxy adhesive (Standard Araldite®) was used to bond an aluminum stud (Ø 15 mm) to the plated samples and a tensile test machine (Shimadzu, Kyoto, Japan, EZ Test) applied the pulling force at a crosshead speed of 5 mm/min.

### *2.4. Tribological Behavior*

The tribological behavior, coefficient of friction (CoF) and wear susceptibility, were evaluated at room temperature and 30%–35% relative humidity in a custom made tribometer with a pin-on-disc configuration, using a 10 mm diameter AISI 52100 steel ball in a cone-shaped holder as the pin. NBR, Ni-P and Ni-P-PTFE $35 \times 35$ mm$^2$ plated samples were tested in triplicate, at a linear speed of 0.1 m/s, using 1, 3 and 5 N loads for 11,500 cycles in 6 and 10 mm track radii. The CoF reported in this work is the average of the last 1000 cycles of 3 samples.

## 3. Results

### *3.1. Surface Morphology*

The concentrations of the PTFE dispersions and CTAB in the Ni-P plating bath were varied according to the most reported conditions in literature for Ni-P-PTFE composite coatings on metallic substrates, Table 1. In total, nine different Ni-P-PTFE-coated samples were prepared, together with three control samples, prepared without the use of surfactant. Exemplary SEM observations of the composite coatings produced with 5, 7 and 10 mL/L of PTFE solution without CTAB are displayed in Figure 1. The samples were named Ni-P-PTFE(x_y), where "x" stands for the PTFE bath concentration (5, 7 or 10 mL/L) and "y" indicates the first number of the CTAB bath concentration (200, 500 and 800 mg/L), e.g., Ni-P-PTFE (5_0) corresponds to a sample plated in a bath containing 5 mL/L of PTFE and 0 mg/L of CTAB.

It is clear from Figure 1 that the co-deposition of PTFE is scarce, very localized (see arrows) and the PTFE particles are not embedded but rather seem to lay on the Ni-P coating surface. Furthermore, the higher the concentration of PTFE, the more agglomeration occurs, as seen from the string of clustered PTFE for a concentration of 10 mL/L. It is also possible to conclude that the μm-sized PTFE particles are predisposed to deposit on the borders and edges of the metallic nodules, as reported by Pancrecious et al. [19]. Finally, it is noteworthy to mention that for the highest concentration of PTFE, sedimentation was observed and a milky white PTFE dispersion was deposited on the bottom of the plating vessel.

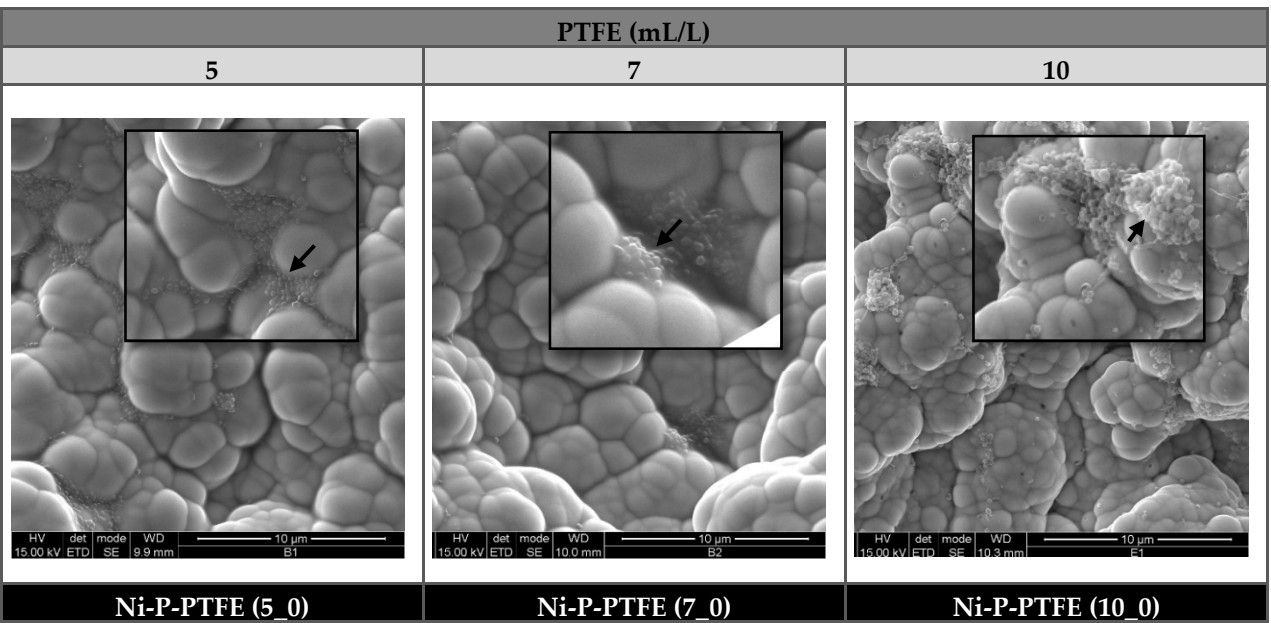

**Figure 1.** SEM micrographs of Ni-P-PTFE coatings on NBR without using surfactant. Arrows indicate PTFE aggregates.

### 3.2. Influence of Cationic Surfactant CTAB

Several authors have studied the influence of surfactants on the particle incorporation in electroless plating coatings and concluded that its success is correlated to the particle's zeta potential and the ability of the surfactant to undergo cathodic reduction [25,30]. The conclusion has been unanimous as to the fact that co-deposition is facilitated when the zeta potential of PTFE particles becomes positive.

Therefore, cationic surfactants are preferred as they adsorb on the PTFE particles creating a positive ionic cloud. The charged particles travel through the bulk of the solution toward the cathodic substrate where they are first loosely adsorbed. Then, if the surfactant undergoes reduction, the chemical process will enable a stronger interaction with the metallic Ni-P matrix [25,31]. CTAB was used in this work as it provides both a positive zeta potential and a cathodic reaction. The alternative are the surfactants of the fluorcarbon-type (FC-type), the most used in literature due to their higher cathodic reactivity [30,32]; however, these are more expensive and they raise serious environmental problems.

The surface of the Ni-P-PTFE coatings deposited with PTFE and CTAB were evaluated as to the dispersion and amount of incorporated PTFE particles; see Figure 2 for the SEM micrographs with the arrows pointing to PTFE particles. It is apparent that a homogeneous particle dispersion is attained for all samples except for Ni-P-PTFE (10_2), corresponding to the highest PTFE concentration with the lowest CTAB concentration.

In this case, the particles are agglomerated in strings comparable with those observed on sample Ni-P-PTFE (10_0) in Figure 1, most likely because the surfactant concentration is not enough to stabilize all PTFE particles, resulting in the sedimentation of PTFE particles. This trend was also found in the literature, namely, in the published work of Zhao et al. [26], who noticed that when using a fluorinated cationic surfactant, FC4, at 500 mg/L, the critical concentration of PTFE before agglomeration and settlement was 24 mL/L. In this work, the highest amounts of well-dispersed surface deposited PTFE particles are observed for samples Ni-P-PTFE (5_2) and Ni-P-PTFE (7_8).

On the other hand, the nodular morphology of the Ni-P coating is not altered by addition of PTFE and CTAB, but the size of the nodules decreases with the amount of CTAB, for the same PTFE concentration, proving that PTFE particles hinder the Ni-P nodules growing process. This is noticeable for the samples plated with 5 and 7 mL/L of PTFE, particularly when one compares the two extreme CTAB concentrations (200 and

800 mg/L) (see Figure 2). The opposite was noticed by Mafi et al. [24] who plated steel disks with 0.7 mL/L of PTFE and CTAB at 100–500 mg/L. Their observation was that high concentrations of surfactant lead to larger Ni-P nodules and a smoother surface. However, Sudagar [28] and Lee [33] uncovered that CTAB has a decreasing effect on the grain size of electroless Ni-P coatings, just like in this work.

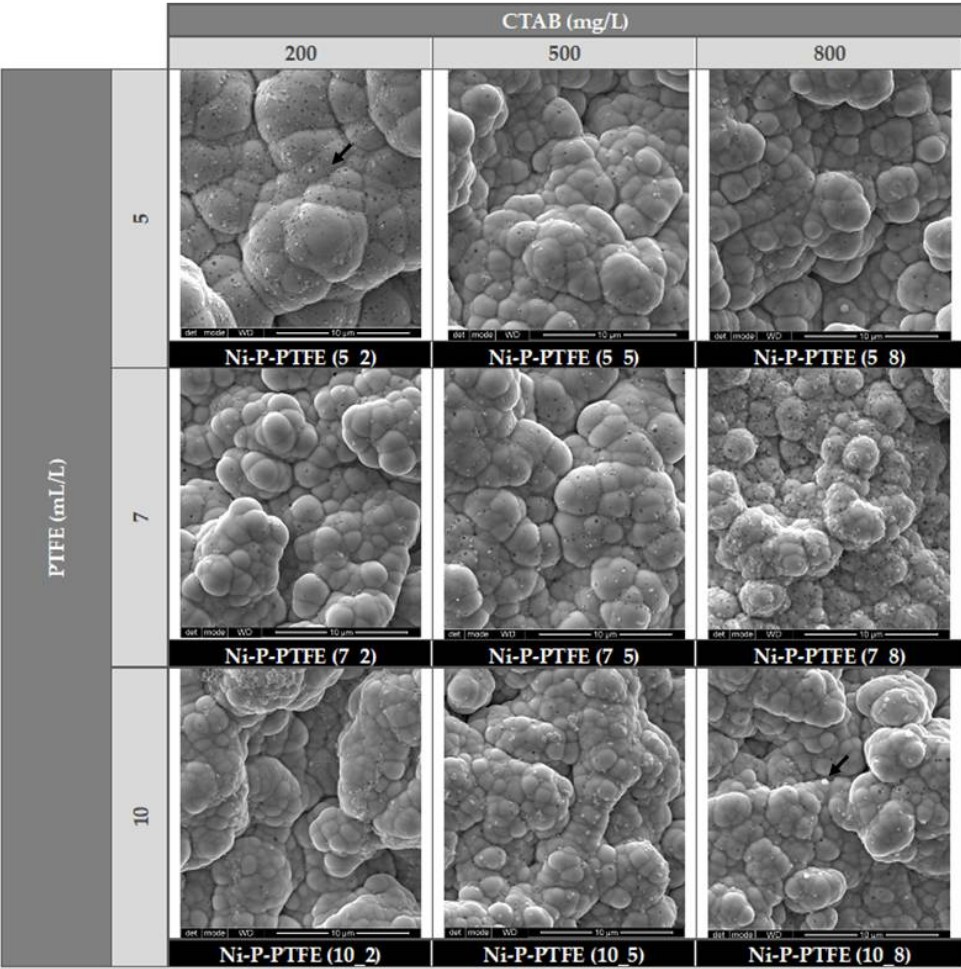

**Figure 2.** SEM micrographs of Ni-P-PTFE samples on NBR varying CTAB and PTFE concentration. The designation of each sample is written bellow its corresponding micrograph. Arrows indicate PTFE particles.

### 3.3. Coating Composition and Plating Rate

To assess the PTFE content in the coatings, EDS analysis was performed simultaneously with the SEM observations. In Figure 3 the atomic percentages (% at) of Ni, P and F in the coatings were calculated. The atomic ratio of Ni/P in the control sample (no PTFE) is 7.1, which gives weight percentages around 90.9 and 6.5 for Ni and P, respectively, allowing us to classify the Ni-P film as a medium-phosphorous coating [14,34]. The composite coatings were compared as to the atomic percentage of F, defined as $F/(Ni + P + F)$, see Figure 3. The results for the F incorporation in electrolessly deposited Ni-P widely vary in the literature, ranging from 30% reported by Zhao et al. [26] to 3% of Yanhai et al. [17,22] for PTFE bath concentrations of 15 mL/L and 12 mL/L, respectively. The results of this work are in the range of those reported by Ger et al. [25,30]. It is important to note that in all cases, these authors used fluorinated alkyl surfactants.

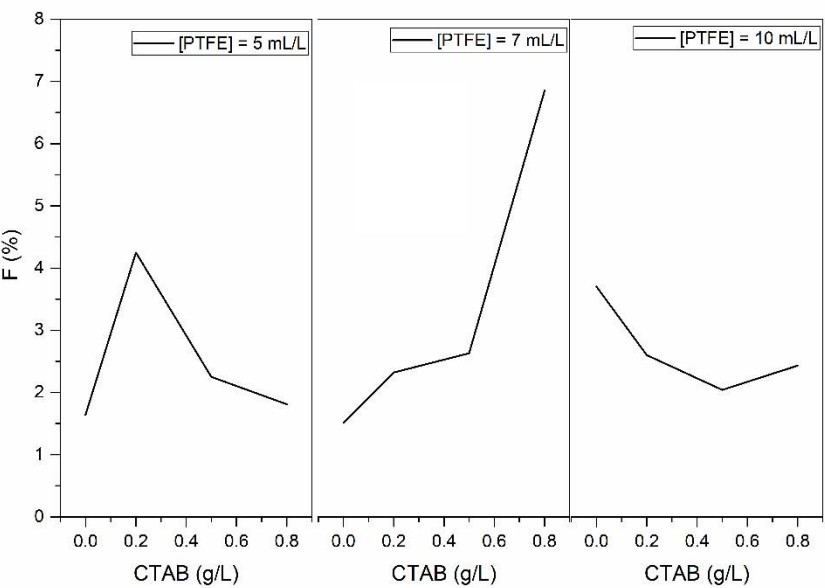

**Figure 3.** Fluorine content in the Ni-P-PTFE films presented in Figures 1 and 2.

From the results of the present work, it can be concluded that the surfactant concentration has opposite effects when the PTFE content in the bath is increased, i.e., at the lowest PTFE concentration of 5 mL/L, the atomic percentage of F decreases with the increase in the CTAB concentration, with a maximum of 4% F for 200 mg/L CTAB, but the inverse trend is observed for 7 mL/L of PTFE, where the maximum incorporation occurs for 800 mg/L CTAB (6.8%). As already mentioned, the CTAB has the effect of adsorbing on PTFE particles, altering zeta potential to positive values and favoring PTFE migration to the cathode. However, if the of PTFE concentration is low (2 mL/L) compared to that of CTAB, the excess of CTAB will adsorb on the Ni-P film, hindering the access of PTFE, an effect that increases with the CTAB concentration.

For the 7 mL/L PTFE concentration, more PTFE-coated particles will be available to migrate to the cathode, particularly for the highest CTAB concentrations 500 and 800 mg/L, leading to an increase in the PTFE particles incorporation in the Ni-P film. Eventually, for the 7mL/L PTFE concentration there is still the possibility to further increase the amount of incorporated F by increasing the CTAB concentration. On the other hand, the cathode coverage with PTFE will hamper the access of $Ni^{2+}$ and $H_2PO_2^{-}$ to the surface, leading to the decrease in the plating rate [25,30].

For the 10 mL/L PTFE concentration, the zero concentration of CTAB corresponds to a large amount of co-deposited PTFE, but it is unevenly distributed, as observed in Figure 1. For this high PTFE concentration, the amount of incorporated F remains low and insensitive to the amount of added CTAB, which may be related to an excess of PTFE that deposits on the Ni-P surface, blocking the deposition sites and further decreasing the composite deposition rate [25,34] (see Figures 1 and 2).

### 3.4. Crystalline Structure and Adhesion

The XRD pattern of the Ni-P and Ni-P-PTFE coatings on rPDA-NBR plated samples is seen in Figure 4. A single, broad peak centered at 2θ = 45° can be observed for all samples, denoting a structure with a low crystallite size. Indeed, Ni presents a peak at about 45° corresponding to the (111) plane of the face-centered cubic phase [24,35,36].

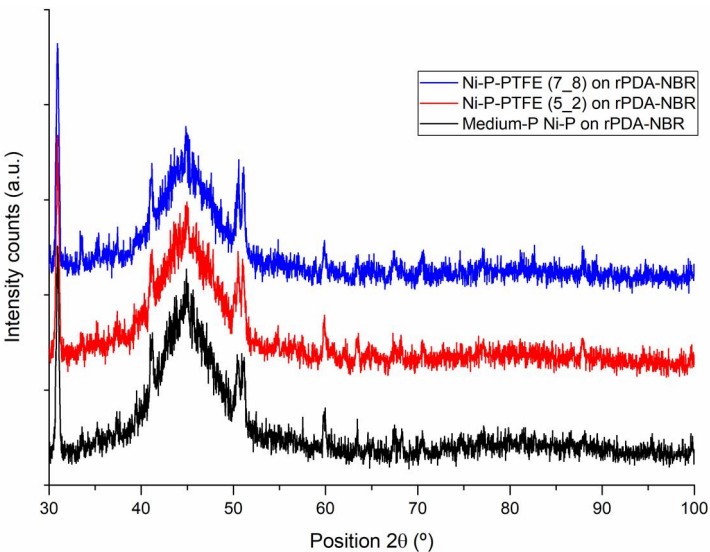

**Figure 4.** XRD analysis of Ni-P, Ni-P-PTFE (5_2) and Ni-P-PTFE (7_8) films.

The XRD patterns are essentially superimposable, meaning that PTFE embedding in the coatings does not lead to alterations in the crystallization behavior of Ni-P, regardless of its content [35].

The adhesion of the films measured by the DPO method was found to be 20% higher for Ni-P-PTFE coatings than for Ni-P coatings, Table 2. A single factor ANOVA analysis ($p < 0.05$) revealed that the adhesion difference between the Ni-P coating and Ni-P-PTFE (5_2) and Ni-P-PTFE (7_8) coatings are statistically significant but not between the PTFE composite coatings.

To explain these results, one must first consider that in this work, the pre-deposition of a thin layer of Ni-P on NBR was performed before starting the PTFE co-deposition, in order to insure a strong interfacial adhesion between the coating and NBR. If this was not performed, the adhesion to NBR would have been weaker than that of Ni-P, as observed by many authors, due to the known PTFE inability to establish a bond with other materials [22,30,37]. Further, the improved adhesion may also be related with the fact that PTFE, as a soft material, enables the dissipation of the internal compressive stress that are present in medium-phosphorous Ni-P coatings [11]. The release of internal stress in electroless Ni-P coatings is known to be associated with increased fatigue strength, reduction in cracks and fissures [38,39] and higher ductility [40] and, as a consequence, higher peeling resistance. Poor Ni-P-PTFE coatings adhesion has also been attributed to hardness mismatch between the metallic substrate and the soft PTFE particles [17,41].

**Table 2.** Adhesion of the Ni-P and Ni-PTFE films measured by the DPO method.

| Parameter | Ni-P | Ni-P-PTFE (5_2) | Ni-P-PTFE (7_8) |
|---|---|---|---|
| Adhesion (MPa) | $1.3 \pm 0.1$ | $1.6 \pm 0.2$ | $1.5 \pm 0.1$ |

*3.5. Roughness and Wettability*

Roughness and wettability of the composite Ni-P-PTFE coatings was assessed for their implications in wear and friction. In addition, a change in the wettability of the coatings is an indirect indicator of the presence of PTFE for its high hydrophobicity.

The wettability results show that PTFE strongly influences the surface properties of the coating, see Table 3. A significant increase in the water CA of ~40° from Ni-P to Ni-P-PTFE surface can be explained considering that in PTFE molecules, the individual fluorine has a very strong electronegativity, easily attracting carbon electrons to form a stable chemical bond [42]. Therefore, the PTFE molecule cannot form hydrogen bonds with

the polar water molecules, revealing a hydrophobic nature that is induced in the Ni-P-PTFE composite coatings.

It is important to note that roughness values are higher than those usually reported for Ni-P based films deposited on metal substrates [20,41]. This is due to the chemical activation treatment underwent by NBR before electroless deposition [15]. On the other hand, roughness decreases from Ni-P to Ni-P-PTFE (5_2), but it increases for Ni-P-PTFE (7_8), where a higher concentration of PTFE particles are incorporated, corroborating the results of You [43].

**Table 3.** Centre line average roughness ($R_a$) and contact angle of Ni-P and Ni-P-PTFE films.

| Parameter | Ni-P | Ni-P-PTFE (5_2) | Ni-P-PTFE (7_8) |
|-----------|------|------------------|------------------|
| Ra (µm)   | 3.15 | 2.41 | 4.20 |
| C.A. (°)  | 70 ± 6 | 136 ± 4 | 131 ± 5 |

### 3.6. Frictional Behavior

The most accepted explanation for the PTFE lubricating effect is that PTFE particles, besides the innate inability to bond to other materials, will soften and distribute under compressive loads, forming a lubricant layer. In addition, its catenarian structure of 13–15 repeated units, combined with the absence of branches and bulkiness, facilitates the deformation of the housing cavities with the Ni-P matrix, squeezing out PTFE particles under pressure that form a thin lubricant film transferred to the counterbody [44].

Figure 5 displays exemplary curves of the CoF evolution for the Ni-P and Ni-P-PTFE coatings with the composites displaying the highest incorporations of PTFE, namely Ni-P-PTFE (5_2) and Ni-P-PTFE (7_8). The bar graph in the bottom right of the Figure summarizes the results in terms of the average CoF at the tested loads of 1 N (0.6 MPa), 3 N (0.9 MPa) and 5 N (1.0 MPa), corresponding to the last 1000 cycles of three samples.

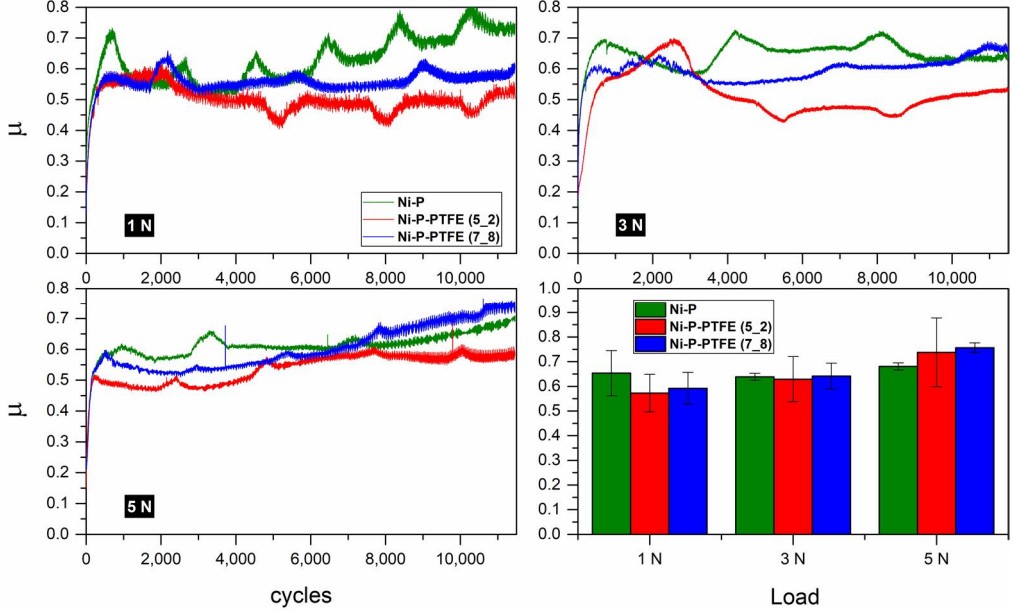

**Figure 5.** Tribological evolution of the Ni-P and Ni-P-PTFE coatings at 1, 3 and 5 N loads (0.6, 0.9 and 1.0 MPa) and average CoF measured for 3 samples for the last 1000 cycles of pin-on-disc tests carried out for 11,500 cycles.

It is apparent that the running-in period is shorter, and the CoF reaches smaller values for the Ni-P-PTFE films than for Ni-P. Since the running-in corresponds to the period of mechanical adjustment of the surfaces to each other, which involves friction and wear, the presence of a conformable and self-lubricating material such as PTFE may speed-up the

process and decrease the friction between surfaces, with an impact in the running-in time and CoF as reported.

On the other hand, the CoF evolution shows some cyclic fluctuations with a periodicity of ~2000 cycles, which appear damped and more spaced with the incorporation of PTFE particles. Such cyclic behavior may be related with the wear and release of particles, leading to a CoF increase, and then the particles are swept away from the surface, resulting in a decrease in the CoF, before the process restarts.

At 1 N (0.6 MPa), the films containing PTFE present a CoF decrease of 9%–12% compared to that of Ni-P, an effect not as expressive as that reported by other authors for the same kind of coatings on metal substrates [32,37]. For higher loads, the lubricant effect of PTFE becomes attenuated, corresponding to a CoF increase with the number of cycles. For example, at 5 N, the coatings containing PTFE go through a stage where the CoF is lower than that of Ni-P, for a low number of cycles, and then the CoF steadily increases over time, and after 11,500 cycles, the CoF for 3 and 5 N becomes indistinguishable of that of Ni-P ($p < 0.05$) (see Figure 5, bottom right).

Straffelini et al. [45] identified two stages in their composite coatings embedding different particles, including PTFE, and considered the end of the first stage to correspond to the surface durability in the tribological testing conditions. Therefore, the transition from stage I to stage II corresponded to a substantial increase in the CoF, which was not observed in this work. The CoF values of the PTFE-containing coatings reported in the literature widely vary in the range 0.15–0.3 [32,37,45–48] to values of the order of 0.6–0.8, these ones being less frequent. For example, Wu et al. [44] plated mild carbon steel with a Ni-P-PTFE film that displayed a CoF of ~0.2–0.3 up to 300 cycles (1 N load), followed by a dramatic increase up to 0.80. Ramalho et al. [49] coated high-speed steel with Ni-P-PTFE and found a CoF of 0.83, higher than that of Ni-P (0.62), which decreased to 0.70 after heat treatment (heat-treated Ni-P had a CoF of 0.53), tested at 2–5 N. The above-mentioned author's explanation for these values was that, due to a low adhesion between film and substrate, the debris particles would be dragged along the surface effecting in a higher CoF, and eventually, the counterbody would reach the substrate by wearing of the films.

However, a direct comparison is practically impossible to establish, as Ni-P-PTFE electroless plating on elastomeric, or on any polymeric material, has never been reported; additionally, thermal treatments, which are known to reduce CoF and enhance coating adhesion, are not possible with a rubber substrate either. However, the comprehensive study that is available in the literature for Ni-P-PTFE coatings on metallic substrates is still a valuable tool to ascertain the plating conditions and F incorporation parameters.

*3.7. Wear Effects*

The wear tracks of Ni-P and Ni-P-PTFE films were observed under SEM, after pin-on-disc tests (11,500 cycles) at 1 and 5 N (see Figure 6). At 1 N, the wear tracks are barely noticeable, whereas for 5 N, they are larger and more distinct from the unworn film, regardless of its composition (see Figure 6 for Ni-P-PTFE(5_2)). The differences in track wideness (see dashed lines) are merely due to higher loadings leading to a larger contact area between counterbody and tested (conformable) substrate, and are not conditioned by the films' composition. To note that the Ni-P-PTFE (5_2) displays a less cracked and smoother surface than Ni-P and Ni-P-PTFE(7_8) films, particularly at 5 N in a difference that may be ascribed, in the case of Ni-P, to the lubricating effect of the PTFE film (arrows indicate fractures); in the case of Ni-P-PTFE(7_8), the difference may be attributed to the higher roughness and higher PTFE content of this film (please compare the cracks indicated by the arrows).

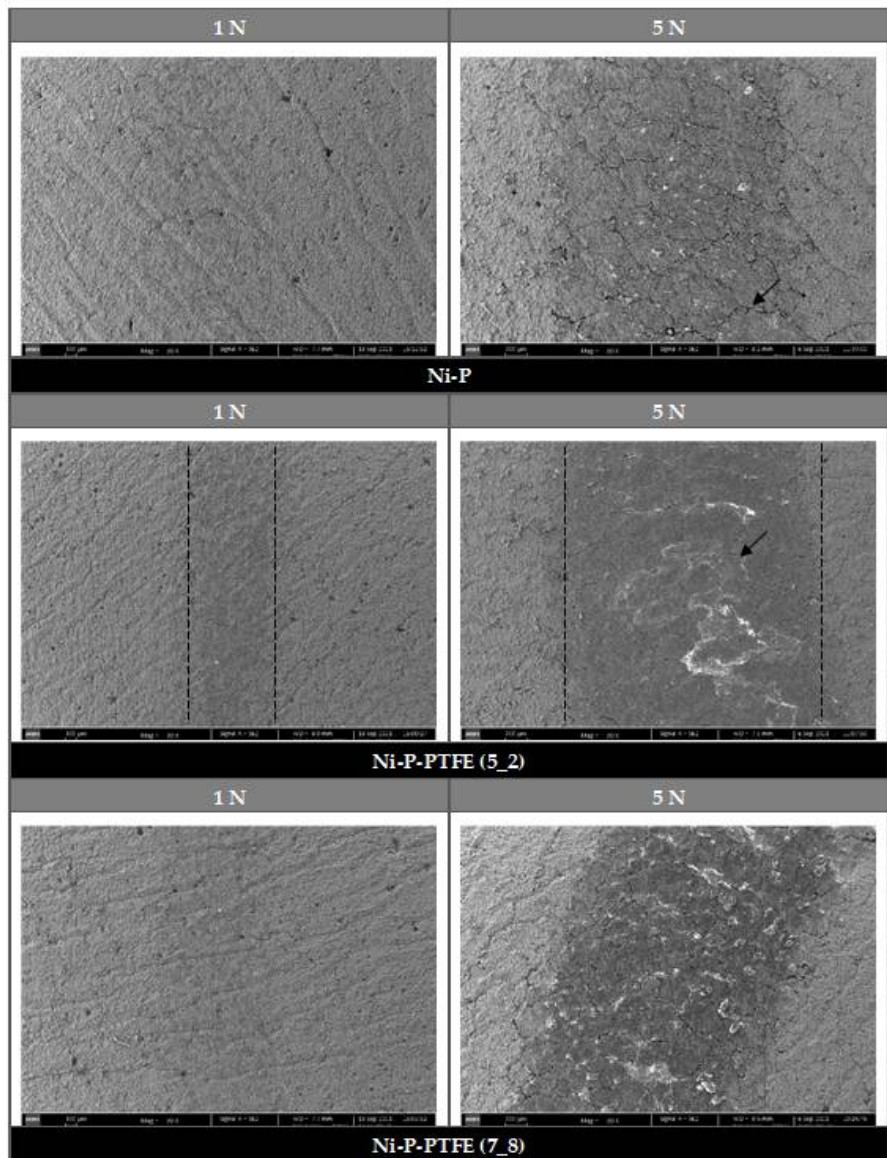

**Figure 6.** SEM micrographs of wear tracks of Ni-P and Ni-P-PTFE films on rPDA-NBR at 30×
magnification. Arrows indicate cracks.

Observing the same tracks at a higher magnification (Figure 7) one can ascertain that
the differences between films go beyond the cracking of the surface. The Ni-P films appear
to have a higher volume of debris and areas where some parts of the film were partially
ripped out as there seems to be a height difference and more empty spaces. The parts that
are more elevated are severely eroded and the nodular structure is absent, whereas the
lower parts show the structure of the original films. It is thus possible that these patches of
elevated and smashed film prevent the steel ball from penetrating further until they are
completed eroded. In these conditions, the CoF value only presented a marginal decrease
compared to that of Ni-P-coated NBR (10%), but the wear damage observed for the PTFE
based coating after 11,500 cycles (1–5 N) was considerably lower than that observed for
Ni-P coatings.

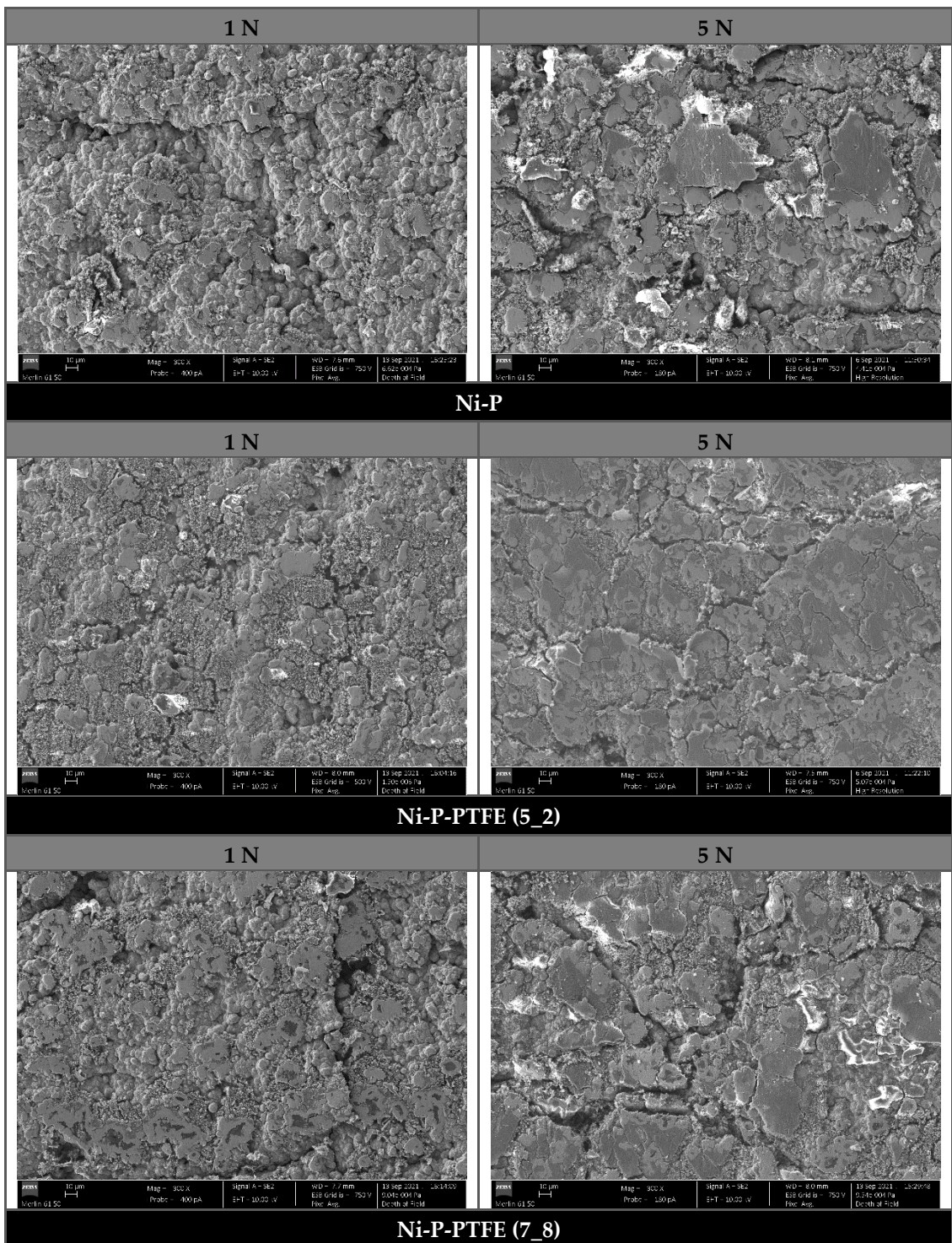

**Figure 7.** SEM micrographs of wear tacks of Ni-P and Ni-P-PTFE films on rPDA-NBR at 300×
magnification.

The Ni-P-PTFE films exhibit a smoother and even, although polished, worn surface,
confirming the observations from the lower magnification images, Figure 6. In particular,
the Ni-P-PTFE(5_2) film displays the least visible cracks and the less worn surface at both
1 and 5 N. Looking at the micrographs from testing at 1 N, it is now understandable that
the fluctuations of CoF observed in Figure 5 were attenuated with PTFE, as its presence
produces smoother, even and less worn surfaces, and therefore, the true contact area
between the two bodies in sliding motion is less variant.

Regardless of the film's composition, the SEM micrographs in Figure 7 suggest that the wearing mechanism is mainly abrasive, contrary to the low-phosphorous Ni-P films of a previous work of the authors [15], where adhesive wear was present as well. This is supported by the fact that there is an absence of grooves and only signs of smearing. For Ni-P-PTFE composite films on metallic substrates, the lack of adhesive wear is attributed to the low surface energy of the PTFE particles, which decrease the true contact area between the counterbody and the tested specimen [14,34,44,50]. Wu et al. [44] further explained that the chemical inertness of PTFE improves the seizure resistance of the Ni-P matrix as a PTFE-rich mechanically mixed layer (PRMML) is formed while the surface is being worn.

Additionally, the non-sticking PRMML decreases the direct contact area between the sliding parts improving the wear resistance. An indirect proof of this phenomenon may be found in this work in the fact that no traces of iron atoms from the steel ball were found in the EDS analysis of the wearing tracks in Figure 8, meaning that adhesive wearing was prevented with the incorporation of PTFE.

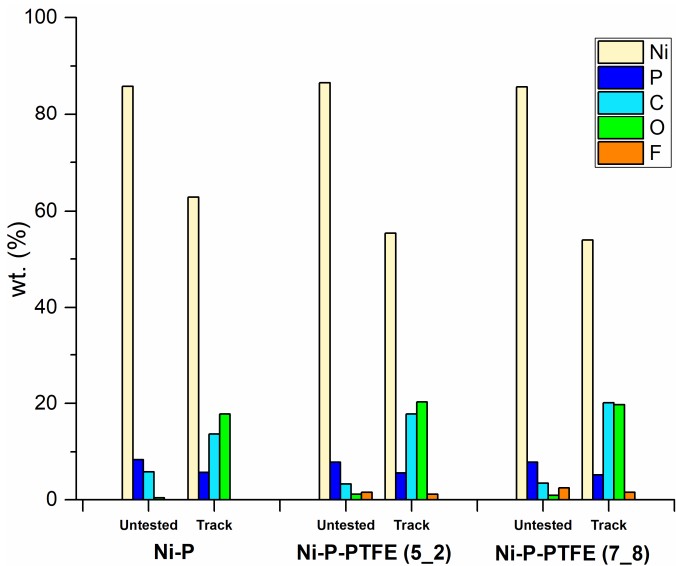

**Figure 8.** Elemental composition in wt. % of an untested area and wear track of Ni-P and Ni-P-PTFE films on rPDA-NBR.

The chemical composition of the wear tracks assessed via EDS analysis also permits us to better understand the effect of wearing during sliding motion in the coatings' composition. For example, the phosphorous content decreases from around 8 wt. % to 5 wt. %. This behavior is unrelated to PTFE and is due to the amorphous nature of medium-phosphorous films which are reported to have a "liquid-like" disorder of the atoms and random distribution of phosphorous in the coatings [51]. The incorporation of phosphorous into the lattice induces a microstrain in the crystalline phase [51] that is alleviated with the compression from the steel ball, exposing the phosphorous atoms to wear. The increase in oxygen is explained by the air oxidation of the films under the effect of friction generated heat, whereas the decrease in nickel is due to wear loss, in the form of powder-like debris, and detachment of the film from the rubber surface. The appearance of fractures can also contribute to the appearance of higher amounts of carbon and oxygen, due to the exposed rubber inside those fractures.

Interestingly, Ramalho et al. [49] had already proved as well that there is a reduction in wear of Ni-P-PTFE composite coatings on steel, although the same coatings were ineffective on the CoF reduction. It is possible that the roughness attributed by the PTFE particles counteracts its lubricant nature as well as that attributed to phosphorous [49]. It is also important to note that in this study, an increase in roughness was observed due to PTFE particles incorporation (see Table 3), which may be correlated to wear. The rare combination

of opposite grades of wear and friction was also found in another study by Wang et al. [52]. In their case, a Ni-P-CNT (carbon nanotubes) composite exhibited a higher wear rate than Ni-P but lower friction due to conglomeration of CNTs, past a certain concentration of the nanowires.

As a matter of fact, even though friction and wear are related to each other as two kinds of responses of a tribosystem, their relationship is not expected to be simple [53]. The incorporation of soft particles in a hard matrix gives new mechanical properties which influence the tribological response, where the behavior of elastomers alone is already erratic.

Looking at the composition of the wear tracks of the Ni-P-PTFE composite films, the wt. % of fluorine atom is decreased by 27% for Ni-P-PTFE (5_2) and 38% in Ni-P-PTFE (7_8), hinting at the fact that rougher PTFE films lead to a worsening of the wear resistance. Moreover, the Ni-P-PTFE (7_8) film displays the highest amount of PTFE; as PTFE is not tightly bonded to the Ni-P matrix, a higher concentration of PTFE will expectedly lead to a mechanically weaker Ni-P-PTFE structure, hence less resistant to wear [54]. These explanations also corroborate the observations of Figures 6 and 7, where the Ni-P-PTFE (7_8) film is superficially more damaged than Ni-P-PTFE (5_2).

Rahmati et al. [38] reported the same decrease in the wt.% of fluorine atoms in the worn surfaces of Ni-P-PTFE plated steel discs and declining wear behavior in films with more PTFE incorporated. Similarly, the authors' opinion was that more PTFE in the films leads to poorer adhesion between the nanoparticles and the nickel matrix.

The wear mechanism reported for PTFE composite films on metallic substrates is similar, namely, being classified as mildly abrasive, displaying a smooth surface with debris and micro cracks [36,37,44,47,49,54]. However, the presence of holes in the neighborhoods of the removed PTFE particles, spalling and film lifted on the edge of the wear track (as reported by these authors) was not observed in the films studied here. That is a natural consequence of the better adhesion to the substrate measured and presented in Table 2.

## 4. Conclusions

In this work, and for the first time, a well-adherent and wear-resistant Ni-P-PTFE coating was successfully deposited on a NBR substrate, by using the electroless plating technique. It was concluded that the presence of CTAB is essential to obtain an homogeneous incorporation of PTFE within the Ni-P plated film, but the concentrations of PTFE and CTAB have to be adjusted in order maximize the amount of co-deposited PTFE.

The two Ni-P-PTFE coatings displaying the highest F concentrations, 4% and 6.8%, were selected for further studies. Both displayed an adhesion force to NBR 20% higher than that of Ni-P films, and an essentially amorphous structure, in line with the structure of Ni-P. The presence of PTFE translated in an increase in the contact angle from 70° to about 130° and a roughness increase in the case of the highest PTFE concentration composite.

The tribological measurements proved that the CoF of Ni-P-PTFE films at 1 N is in the order of 0.56–0.58, about 9%–12% lower than that of Ni-P (CoF = 0.65), and for higher loads, no differences were detected related with the presence of PTFE.

More notorious effects of the presence of PTFE in the Ni-P coatings were noticed regarding the wear effects, with the Ni-P-PTFE coatings displaying a smoother surface. Additionally, the PTFE containing films, and particularly the Ni-P-PTFE(5_2) film, display a less damaged surface after the tribotests, with a lower volume of debris than the Ni-P film after 11,500 cycles, clearly demonstrating the effectiveness of the PTFE presence in slowing down the wear effects with loads in the range 1–5 N.

In conclusion, the Ni-P-PTFE coatings may be a cost-effective alternative solution to the DLC films, particularly for those applications where the lowest CoFs (e.g., 0.1–0.3) are not needed.

**Author Contributions:** Conceptualization, B.V.; methodology, B.V. and R.S.; validation, R.S., J.O. and C.F.; writing—original draft, B.V.; writing—review and editing, C.F. and J.O. All authors have read and agreed to the published version of the manuscript.

**Funding:** This research was funded by national funds through Fundação para a Ciência e Tecnologia (FCT) (Grant Nos. PT/BD/128477/2017 and UIDP/00285/2020).

**Institutional Review Board Statement:** Not applicable.

**Informed Consent Statement:** Not applicable.

**Data Availability Statement:** Not applicable.

**Acknowledgments:** The authors would like to acknowledge CEMUP and IPN for SEM analysis.

**Conflicts of Interest:** The authors declare no conflict of interest. The funders had no role in the design of the study, in the collection, analyses, or interpretation of data, in the writing of the man script, or in the decision to publish the results.

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
