# Peer review of "Characterization and Tribological Behavior of Electroless-Deposited Ni-P-PTFE Films on NBR Substrates for Dynamic Contact Applications"

_coatings, doi:10.3390/coatings12101410_

Round 1

Reviewer 1 Report

  1. While using first time, expand NBR in abstract
  2. Why this coating is necesary and what is the purpose of studying tribological behaviour is not convincing. With an application, kindly explain this work necessity
  3. Encircle the inference in SEM images
  4. Range of utilizing PTFE & CTAB selection is not justified
  5. Figure 4 include error bar chart
  6. Ra value seems to be high, Kindly check and discuss the reasons
  7. The basis of selecing low load 1, 3 , 5 N in pin on disc should be justified with respect to application
  8. What is the speed used in tribological tetsing and time?
  9. Why authors mentioned cycles in X axis. 
  10. The specimen size and its image should be included in text
  11. how uniform contact is ensured for low loads here?
  12. What is the plate material and its hardness value?
  13. Fig 7, encircle and convey the findings
  14. Most literatures are too old. try to include last 5 years papers

Author Response

The authors would like to acknowledge the reviewer for the comments that enabled us to significantly improve the quality of the original paper.

Q.1: While using first time, expand NBR in abstract

Answer: NBR was expanded in the abstract.

Q.2: Why this coating is necessary and what is the purpose of studying tribological behaviour is not convincing. With an application, kindly explain this work necessity

Answer: Some examples of potential applications of these films are O-rings for zoom cameras, ball bearings and rubber seals for the aerospace industry, and rural water supply industry, as reported in reference [10].  Some examples of potential applications in the automobile industry can be found in reference [11]. Finally, a paper dedicated to a direct application is given in reference [3]. A sentence was added in the Introduction section providing some application examples (see yellow highlighted text).

 [1] Lubwama, M.; Corcoran, B.; Sayers, K. DLC films deposited on rubber substrates: a review.
Surface Engineering 2015, 31, 1-10, DOI:https://doi.org/10.1179/1743294414Y.0000000379.

[2] Khan, M., Heinrich, G. PTFE-Based Rubber Composites for Tribological Applications, Adv Polym Sci 2011, 31,249-310, DOI: 10.1007/12_2010_98.

[3] Veronesi, P.; Sola, R.; Poli, G. Electroless Ni coatings for the improvement of wear resistance of bearings for lightweight rotary gear pumps. International Journal of Surface Science and Engineering 2008, 2, 190-201, https://doi.org/10.1504/IJSURFSE.2008.020493.

Q.3: Encircle the inference in SEM images

Answer: The authors think the Reviewer refers to the SEM images of Figs.1 and 2. Arrows were added to Figures for clarification, and a reference to the arrows was made in the text and Figure's legends (see highlighted yellow text).

Q.4: Range of utilizing PTFE & CTAB selection is not justified

Answer: The used PTFE and CTAB concentration ranges are based on the concentrations ranges most used in the literature for the electroless plating formation of Ni-P-PTFE composite films. The justification is given in the first paragraph of section 3.1: “The concentrations of the PTFE dispersions and CTAB in the Ni-P plating bath were varied according to the most reported conditions in literature for Ni-P-PTFE composite coatings on metallic substrates

Q.5: Figure 4 include error bar chart

Answer: Thank you for this comment. As the authors didn’t repeat these experiments, error bars cannot be included, and indeed they are needed to draw conclusions because some of the data points are quite close to each other. Owing to the pertinent comment of the Reviewer, the authors decided to remove this Figure. The statement on the influence of CTAB and PTFE concentrations on the plating rate was kept in the text, but supported by two references, see p. 7.

Q.6: Ra value seems to be high, Kindly check and discuss the reasons

Answer: The reviewer is right, common values for Ni-P roughness are in the sub-micron range, for electroless plating films. However, the NBR substrate in this case was first subjected to an activation treatment,  where the NBR underwent a chemical attack that substantially increased the roughness, as mentioned in our previous work (reference 15). A remark on the high Ra value was added to the text, section 3.4, second paragraph, p.9 (see highlighted yellow text).

Q.7: The basis of selecting low load 1, 3 , 5 N in pin on disc should be justified with respect to application

Answer: The used loads were chosen for comparison with DLC coatings applied on NBR and HNBR for the purpose of dynamic contact applications.. Examples can be found in the answer to Q2.

Q.8: What is the speed used in tribological testing and time?

Answer: The testing speed is mentioned in section 2.4 of the experimental part. For this work the authors used 0.1 m/s. The testing time was 120 minutes for 10 mm radii and 72 minutes for 6 mm radii, since the authors wanted to keep the number of cycles constant at 11450.

Q.9: Why authors mentioned cycles in X axis. 

Answer: Due to some limitations in the number of samples the tests were performed using 6 and 10 mm radii, to have the same number of times that the pin passes on the sample surface. For this reason, the testing time was adjusted accordingly and as a result, the distance between the two radii is not the same. For graphic clarity, it is simpler to have the number of cycles in X axis meaning that the authors can directly compare the results.

Q.10: The specimen size and its image should are included in text

Answer: The specimens used for the tribological tests have the dimensions: 35 x 35 mm, as stated in section 2.4.The image of a plated sample was also included in Table 1.

Q.11: how uniform contact is ensured for low loads here?

Answer: The pin-on-disk apparatus is equipped with a damping system to help and maintain uniform contact between the tested sample and pin. In this work, the CoF results give some indication of uniform contact even for the low loads, the results at 1N do have a slightly higher noise but the measured values have a good correlation with the other loads.

Q.12: What is the plate material and its hardness value?

Answer: The pin used was a 10 mm diameter AISI 52100 steel ball (hardness 140 MPa), the sample was the treated NBR sample attached to a stainless steel plate (hardness 400 MPa).

Q.11: Fig 7, encircle and convey the findings

Answer: The first paragraph of section 3.6 and Figure 7 were changed to take into account the comment (see yellow highlighted text).

Q12: Most literatures are too old. Try to include last 5 years papers

Answer: The reviewer is right. The authors replaced three of their old references by more recent works, from 2022, 2021 and 2019 (references 16, 20 and 23 of the manuscript that replaced older references):

Chen, Z.; Zhu, L.; Ren, L.; Liu, J., Chen, Z.; Zhu, L.; Ren, L.; Liu, J., Electroless Plating of Ni-P and Ni-P-PTFE on Micro-Arc Oxidation Coatings for Improved Tribological Performance. Mat. Res. 2022, 25:e20220096, https://doi.org/10.1590/1980-5373-MR-2022-0096.

Li, F.; Yang, Y.; Song, L.; Liang, L. Simultaneous friction and wear reduction of Ni-P/PTFE composites under dry sliding condition. Ind. Lub. and Tribology 2021 73(4), https://doi.org/10.1108/ILT-07-2020-0235.

Hadipour, A.,; Rahsepar, M.; Hayatdavoudi, H. Fabrication and characterisation of functionally graded Ni-P coatings with improved wear and corrosion resistance. Surf. Eng. 2019, 35(10) 883-90, https://doi.org/10.1080/02670844.2018.1539295.

However, some of the papers more than 5 years old still reference papers in the area, at least from the methodology point of view

Reviewer 2 Report

1.       Actually, based on the introduction, the authors are using “(nano)composite” idea to tune the Ni-P tribological performance. I suggest briefly comparing this system with other metal (nano)composites and make the difference and impacts stand out. The literatures below could be a good reference, and I suggest adding them to have a short discussion.

[1] Metal matrix nanocomposites in tribology: Manufacturing, performance, and mechanisms. S Pan, K Jin, T Wang, Z Zhang, L Zheng, N Umehara. Friction, 1-39

[2] Ren, Y., Zhang, L., Xie, G., Li, Z., Chen, H., Gong, H., Xu, W., Guo, D. and Luo, J., 2021. A review on tribology of polymer composite coatings. Friction, 9(3), pp.429-470.

2.       Fig.3 and 4: Why 0.5g/L CTAB has a different trend for plating rate? Must explain from fundamental point of view…this is not right for now.

3.       Change Fig. 3 to 2D stacking figure. Current 3D plot is NOT scientific or easy for comparison.

4.       For tribological tests, also use MPa instead of N for analysis. E.g., how many MPa is it for 1N, 3N, and 5N?

5.       Fig. 6: Based on 1-5N results, the running-in period for Ni-P-PTFE seems to be always better. The author should divide the running-in period and the stable period, and then discuss why the stable period shows higher CoF in Ni-P-PTFE.

6.       Mark the wear cracks in Fig.7 & 8

7.       Fig.9: Do not need to show Ni. Mark as “bal.”

8.       Fig.1 and 2 shows so many different samples. Why only choose 5_2 and 7_8 for tribological study? At least, the authors MUST add one 10_x sample for tribological comparison.

Author Response

The authors would like to acknowledge the reviewer for the comments that enabled us to significantly improve the quality of the original paper.

Q.1: Actually, based on the introduction, the authors are using “(nano)composite” idea to tune the Ni-P tribological performance. I suggest briefly comparing this system with other metal (nano)composites and make the difference and impacts stand out. The literatures below could be a good reference, and I suggest adding them to have a short discussion.

[1] Metal matrix nanocomposites in tribology: Manufacturing, performance, and mechanisms. S Pan, K Jin, T Wang, Z Zhang, L Zheng, N Umehara. Friction, 1-39

[2] Ren, Y., Zhang, L., Xie, G., Li, Z., Chen, H., Gong, H., Xu, W., Guo, D. and Luo, J., 2021. A review on tribology of polymer composite coatings. Friction, 9(3), pp.429-470.

Answer: Thank you for the comment. A small discussion of the types of nanoparticles that may be incorporated to reduce friction and wear was introduced, justifying the choice for PTFE, as suggested ( p.2, third paragraph). The authors appreciated and read the references suggested by the Reviewer, but they had already included a reference with information similar to those suggested by the Reviewer, namely:

Sudagar, J.; Lian, J.; Sha, W. Electroless nickel, alloy, composite and nano coatings–A critical review. J. Alloys Compd. 2013, 571, 554 183-204, https://doi.org/10.1016/j.jallcom.2013.03.107.

Q.2: Fig.3 and 4: Why 0.5g/L CTAB has a different trend for plating rate? Must explain from fundamental point of view…this is not right for now.

Answer: Another Reviewer also questioned the validity of the data in Fig.4 as no error bars were added. Since these experiments were not repeated, the authors decided to remove this Figure from the paper. The statement on the influence of CTAB and PTFE concentrations on the plating rate was kept in the text, but supported by two references, see p. 7.

Q.3: Change Fig. 3 to 2D stacking figure. Current 3D plot is NOT scientific or easy for comparison.

Answer: The figure was redesigned according to the indications of the Reviewer.

Q.4:  For tribological tests, also use MPa instead of N for analysis. E.g., how many MPa is it for 1N, 3N, and 5N?

Answer: The contact forces during pin-on-disk testing of this work were: 0.6, 0.9 and 1.0 MPa for loads of 1, 3 and 5 N respectively. This clarification was added to the text, first paragraph of p.10 and legend of Figure 6.

Q.5: Fig. 6: Based on 1-5N results, the running-in period for Ni-P-PTFE seems to be always better. The author should divide the running-in period and the stable period, and then discuss why the stable period shows higher CoF in Ni-P-PTFE.

Answer: The existence of the running-in period, its extension, and associated CoF, are now explained in the text, please see the first paragraph of page 10.

Q.6: Mark the wear cracks in Fig.7 & 8

Answer: As requested, examples of cracks are indicated in Figures 7 and 8 (now 6 and 7) and this is mentioned in the text, please see Figures 6 and 7 and section 3.6.

Q.7: Fig.9: Do not need to show Ni. Mark as “bal.”

Answer: The authors would like to keep the reference to nickel as the decrease of the bar ascribed to this element is related to wear and it is significant because it decreases during the tribological experiment..

Q.8: Fig.1 and 2 shows so many different samples. Why only choose 5_2 and 7_8 for tribological study? At least, the authors MUST add one 10_x sample for tribological comparison.

Answer: The Ni-P-PTFE(5_2) and Ni-P-PTFE(7_8) were chosen as they showed the highest PTFE incorporations (please see Fig. 2 and Fig.3), while providing an homogeneous dispersion of PTFE in the coating.

 As it can be seen in Fig.2 and confirmed on Fig.3, the Ni-P-PTFE(10_x) samples present lower concentrations of PTFE and aggregated particles. Furthermore, PTFE sedimentation was observed, in the form of a milky white dispersion on the bottom of the plating vessel, as stated in the first paragraph of p.5. This Figure proves that the addition of a surfactant is essential to perform the incorporation in good conditions.

Reviewer 3 Report

The article was properly prepared. It contains interesting research results that should be published. The article may be published in the presented format.

Round 2

Reviewer 1 Report

Nil

Author Response

.

Reviewer 2 Report

As I mentioned, Fig. 3 should use "2D" figures for readability...Why you still use 3D histogram? This can not be used for direct visual comaprison.

Author Response

Dear Reviewer,

Figure 3 results are now presented in the form of a set of 2D Figures.

Thank you

Sincerely,

carlos fonseca